# Early Life Exposure to Food Contaminants and Social Stress as Risk Factor for Metabolic Disorders Occurrence?—An Overview

**DOI:** 10.3390/biom11050687

**Published:** 2021-05-03

**Authors:** Laurence Guzylack-Piriou, Sandrine Ménard

**Affiliations:** 1INRAE, 31076 Toulouse, France; 2IRSD, Université de Toulouse, INSERM, INRAE, ENVT, UPS, 31024 Toulouse, France; sandrine.menard@inrae.fr

**Keywords:** food contaminants, social stress, metabolic diseases

## Abstract

The global prevalence of obesity has been increasing in recent years and is now the major public health challenge worldwide. While the risks of developing metabolic disorders (MD) including obesity and type 2 diabetes (T2D) have been historically thought to be essentially driven by increased caloric intake and lack of exercise, this is insufficient to account for the observed changes in disease trends. Based on human epidemiological and pre-clinical experimental studies, this overview questioned the role of non-nutritional components as contributors to the epidemic of MD with a special emphasis on food contaminants and social stress. This overview examines the impact of early life adverse events (ELAE) focusing on exposures to food contaminants or social stress on weight gain and T2D occurrence in the offspring and explores potential mechanisms leading to MD in adulthood. Indeed, summing up data on both ELAE models in parallel allowed us to identify common patterns that appear worthwhile to study in MD etiology. This overview provides some evidence of a link between ELAE-induced intestinal barrier disruption, inflammation, epigenetic modifications, and the occurrence of MD. This overview sums up evidence that MD could have developmental origins and that ELAE are risk factors for MD at adulthood independently of nutritional status.

## 1. Introduction

In recent decades, metabolic disorders (MD) encompassing obesity, type 2 diabetes, and cardiovascular diseases have been increasing worldwide. Indeed, the World Health Organization (WHO) claimed that obesity has nearly tripled worldwide since 1975. In 2016, 39% of adults were overweight, and 13% were obese. Most of the world’s population live in countries where overweight and obesity kill more people than underweight. The consequence of overweight and associated diabetes is the cardiovascular risk (https://www.who.int/news-room/fact-sheets/detail/obesity-and-overweight, accessed on 15 January 2021). In adults, obesity and type 2 Diabetes (T2D) are multifactorial diseases that, in most cases except for genetic conditions (MCR4 mutations, Prader–Willi, Bardet–Biedl …), results from an imbalance between energy intake and expenditure.

MD are multifactorial diseases with exogenous and endogenous origins. They are tightly linked to way of life, specially food intake and sedentary lifestyle. Interestingly, T2D and obesity are associated with defects of intestinal barrier functions (microbiota, permeability, gut-associated immune system) and systemic inflammation. This observation questioned the potential role of environmental stressors despite nutritional factors known to impair intestinal barrier functions and systemic inflammation in triggering metabolic disorders.

The rise of MD in industrialized countries over the past few decades has been paralleled with changes in the handling, medical care, and diet of pregnant mothers, neonates, and infants during their first years of life. This has led to the concept of the “neonatal window of opportunity”, considering the postnatal period as a non-redundant time period after birth during which microbial, nutritional, and environmental exposure prime the host’s organ development as well as innate and adaptive immune system. This issue should be carefully addressed, as highlighted by the mutual report from WHO, UNICEF, and Lancet commissions [1].

The Developmental Origin of Health and Diseases (DOHaD) [2] is a concept that might contribute to the development of MD. Indeed, pre- and postnatal periods represent a critical time window of maturation for the organism (immune system, intestinal barrier, and nutritional switch). Complementary data coming from epidemiological studies and experimental animal studies suggest that exogenous factors during perinatal life influence the life-long susceptibility to MD in adulthood independently of nutritional status. The aim of this overview is to compile evidence highlighting a relationship between ELAE-induced (food contaminants or social stress) intestinal barrier disruption, inflammation, epigenetics, and the occurrence of MD. Our goal is to combine evidences based on a review of the literature and offer a new field of research and perspectives on the role of DOHaD in MD occurrence.

## 2. Early Origins of Metabolic Disorders

In mice and humans, early life is important for the development of the immune system, metabolic switch, microbiota colonization [3], and the development of life-long beneficial host-microbe homeostasis [4]. The first study linking early life adverse events and MD is based on the data from the Dutch famine (1944–1945). Data from this cohort demonstrated that famine during early gestation led to heavier babies at risk of developing obesity and/or cardiovascular diseases in adulthood, whereas famine during middle or late gestation led to low birth weight (LBW) babies with a high risk of impair glucose tolerance in adulthood [5]. These data highlight the important notion of time of exposure. Indeed, the same early life adverse events can have different consequences depending on the window of exposure. These results were comforted by data from the Chinese famine (1959–1961) that were more statistically powerful due to the higher number of persons affected and the higher duration of the famine. Women exposed to famine as fetuses have a higher risk of metabolic disorders that women exposed postnatally [6], whereas no effect was observed in men. Thus, not only can the gestational stage for exposure to famine affect the outcome, but also the sex of the fetus. In the Nigerian famine, occurring during civil war (1967–1970), fetus exposure to famine was also associated with increased risk of MD [7]. Data from famines were of great interest to show the association between ELAE and risk of developing MD. However, famines are not only characterized by food restriction, but also by the stressful environment that appeared as confounding factors.

On the other hand, obesity due to excess of nutrients and/or feeding with a Western diet and lack of physical activity drive women with high Body Mass Index (BMI) to enter the first trimester with MD and/or experience significant gestational weight gain. The American College of Obstetricians and Gynecologists identified maternal obesity as the greatest public health risk in pregnancy and the higher number of large gestational age infants are born from mothers with obesity [8]. The intrauterine environment in obesity is characterized by nutrient excess and associated with an increased risk for childhood obesity [9]. Studies showed that both maternal [10,11,12] and paternal [13] pre-pregnancy high BMI are associated with higher childhood BMI. Furthermore, greater weight gain in the first trimester was linked with risk of overweight in children [14]. The excess of sugar and lipid intake by the fetus/baby contributes to the risk of developing MD in adulthood. However, high BMI in mothers is usually associated with diet; besides nutrients, diet—especially highly processed diets—also contains food contaminants. Furthermore, epidemiological evidence supports not only the need for early-life interventions to reduce the obesity and diabetes burden in later life, but also preconception intervention (for review [15]), highlighting a potential role of epigenetics that is discussed in this overview.

In our societies, diet not only contains nutrients but also food contaminants that could be confounding factors leading to MD at adulthood. The role of early life exposure to food contaminants in MD development is discussed in this overview.

Interestingly, the DOHaD concept was initiated by a study demonstrating an association between low birth weight and risk of ischaemic heart disease in later life [16]. It was clearly demonstrated since the 1990s that early life exposure can imprint the metabolism of the offspring, especially given that maternal and postnatal nutrition are risk factors for the development of T2D [17]. This field of research is of great interest, as a better understanding in behavior affecting MD will provide prevention strategies and the promotion of public health regarding metabolic disorders. Many cohorts in children (TEDDY, HAPO, ABCD, TODAY, PEACHES, EPOCH) addressed the role of early life exposures in T2D occurrence (for review [18]). Most of those cohorts focused on how maternal metabolism (including gestational diabetes mellitus, GDM) and their nutritional status will affect offspring metabolism.

Based on these studies showing that nutritional restriction or excess in early life are associated with outcomes on offspring metabolism, both authors took advantage of their expertise in the field of long-time consequences of early life stressful adverse events and exposure to food contaminants to propose that they could be considered as cofounding risks factors in the development of metabolic disorders.

We developed this hypothesis based on epidemiological and experimental studies that allow working on control models with limited confusing factors and identifying if perinatal exposure to food contaminants or stressful events beside their association with food are risk factors for the development of MD at adulthood. Developing this hypothesis, from two different models of ELAE, appears to be a strength to identify a plausible link between ELAE and MD occurrence and decipher the mechanisms involved.

## 3. Food Contaminants or Social Stress Exposure in Early Life Are Suspected to Contribute to Metabolic Disorders

### 3.1. Role of Environmental Contaminants

Among environmental contaminants, a large focus has been placed on endocrine-disrupting chemicals (EDCs), motivated by their ability to interfere with hormone action and their being widespread. EDCs are exogenous compounds able to disrupt the endocrine system, and thus interfere with organ development, reproduction, immunity, metabolism, and behavior. The effects of EDCs, like those of hormones, can occur at very low levels and follow non-monotonic dose effects (often referred to as biphasic or U-shaped responses).

High concentrations of EDCs are found in several everyday products including plastic bottles and food containers, and the leakage of EDCs into liquid or food content leads to dietary intake (review [19]). Some EDCs are difficult to excrete and can bioaccumulate in tissues. More than 85,000 chemicals are registered in commerce and most of them are poorly studied. Current estimation identifies only approximately 1000 chemicals that meet the criteria of an EDC [20].

In the context of DOHaD, endocrine disruptors were identified as a risk factor in fetal programming. This is facilitated by the ability of EDCs to cross the placental barrier and then to enter in the fetal circulation [21].

EDCs were widely investigated as “obesogenic” factors disrupting both pregnancy and fetal physiology. The obesogen hypothesis highlights two important issues. First, predisposition to obesity starts as early as in the fetal condition and/or the first few years of life. Second, a specific subclass of EDCs known to alter development of fetus and babies could impair weight gain functions later in life. In 2015, the Parma Consensus Statement proposed that the “obesogen hypothesis” should be expanded, based on new data showing that chemicals increased susceptibility to T2D, liver lipid abnormalities, and metabolic disorders (MD) [22]. The Parma statement proposed the name “metabolism disrupting chemical (MDC) hypothesis” to distinguish the role of chemicals from other metabolic disruptors such as nutrition and stress. The MDC postulates that EDCs have the ability to promote metabolic changes leading to obesity, T2D, or fatty liver in animals, including humans; these EDC-induced metabolic alterations may play an important role in the global epidemic of MD. Indeed, there are both in vivo experimental data and epidemiological evidence to support such a hypothesis.

#### 3.1.1. Focus on Endocrine Disruptors Chemicals (EDCs): Bisphenols and Phthalates

Recently, considerable interest has been raised regarding the biological effects of EDCs, particularly in the bisphenol family (BPs) and phthalates. BPs and phthalates are widely present in everyday life due to the considerable volume of plastic produced. They could be released at room temperature, but heating facilitates their leakage, resulting in massive food and beverage contamination [21].

Bisphenol A (BPA) is an industrial chemical that has been used to produce plastics and resins since the 1960s. BPA is found in polycarbonate plastics and epoxy resins commonly used in containers to store food, beverages, and other goods. Early life exposure to BPA has detrimental effects on health and, consequently, the European Union prohibited BPA use—at least in polycarbonate baby bottles—in 2011. BPA was slowly removed and replaced by its structural analogs, Bisphenol-S (BPS) and Bisphenol-F (BPF). Nevertheless, the safety of BPS and BPF is discussed and could mimic detrimental effects of BPA after early life exposure [23]. Metabolism on BPs by the organism are important to consider. BPA is absorbed by the digestive tract [24], metabolized by the intestinal microbiota, and then detoxified by the liver, leading to BPA monoglucuronide (BPA-G) [25,26]. Data on BPS and BPF metabolism are lacking [24].

Phthalates are diesters of phthalic acid classified into high- and low-molecular-weight. High molecular weight phthalates including di (2-ethylhexyl) phthalate (DEHP) have properties to increase plastic flexibility and durability. Low-molecular-weight phthalates, such as diethyl phthalate (DEP), are mainly used in personal care products and cosmetics, but also in pesticides and in food packaging [27].

Lopes Almeida et al. (2019) recently reviewed the role of the environment in MD occurrence [28]. Indeed, prenatal BPA exposure in humans was associated with increased body fat at 7 years old [29] and BMI at 9 years old [30], or accelerated postnatal growth without a change in BMI between 2 and 5 years old [31]. These data are consistent with the DOHaD prediction that light babies at birth would experience increased rates of growth in childhood [32]. However, longitudinal studies of prenatal BPA exposure found either no association with children BMI or a weak non-significant association [33]. Agay-Shay (2015) found no association between BPA or phthalates during pregnancy (urine, serum, or cord blood) and overweight at 7 years old [34]. Special attention to confounding factors is needed; indeed, diet is not the only source of BPA exposure. Moreover, since the developing organism is more susceptible to BPA effects, epidemiology studies must consider the life stage when exposure is measured.

Phthalate exposure during pregnancy was also associated with increased body mass three months after birth in boys [35]. While these data are insufficient to draw clear-cut conclusions, they support the hypothesis that developmental exposure to DEHP, and perhaps other phthalates, can lead to increased weight gain in animals and humans later in life.

#### 3.1.2. EDCs Could Trigger MD in Offspring by Inducing Gestational Diabetes Mellitus (GDM)

Pregnancy is a diabetogenic challenge due to the increased demand for glucose for fetus growth and development [36]. GDM is the most common metabolic dysregulation during pregnancy and affects 5–9% of women in the US [37]. GDM occurs when pregnant women are unable to balance the increased requirements of insulin [38]. A risk factor for GDM is an excessive weight gain during pregnancy. Complications of GDM for the offspring are increased birth weight and predisposition for MD (American Diabetes Association, 2014). Despite GDM, a mother’s pre-pregnancy high BMI was associated with the occurrence of MD in later life, independently of birth or perinatal conditions, socioeconomic characteristics, and health behaviors [39].

Based on various cohorts, many studies investigated a possible association between pregnancy exposure to EDCs and occurrence of GDM; however, the results are not clear-cut. Associations between an increase in BPA concentration in the urine of 1213 pregnant women and lower mid-to late pregnancy gestational weight gain was observed in a population-based prospective cohort study [40]. Differently, in the LIFECODES cohort recruiting 347 pregnant women, early gestational weight gain (median time period: 7.4 gestational weeks) was associated to higher urinary phthalate metabolite concentrations, the mono-ethyl-phthalate (MEP), during the first trimester. The association between MEP and gestational weight gain showed a U-shape, whereas the link between DEHP and gestational weight gain was described by an inverse U-shape [41]. In a Chinese cohort prospective study including 620 pregnant women, the odds of GDM was reduced by 27% for each unit increase in natural log-transformed BPA [42]. On the contrary, in a small case-control study, no evidence of an association between BPA exposure and GDM diagnosis was found [43]. No statistically significant associations were observed between first-trimester urinary BPA concentration with diagnosis of impaired fasting glycemia (IFG) or GDM, even in the Maternal-Infant Research on Environmental Chemicals (MIREC) cohort study [44]. The later study also failed to find an association between urinary concentration of phthalates metabolites and risk of IFG and GDM.

Alonso-Magdalena et al. (2015) showed in animal models that pregnancy is a critical window of susceptibility for BPA effects, potentially causing glucose intolerance and altered insulin sensitivity in mothers later in their lives [45].

In the LIFECODES cohort, exposure to MEP during the second trimester of pregnancy was associated with increased odds of Impaired Glucose Tolerance (IGT) and DEHP exposure was inversely associated with IGT. In both cases, the confidence intervals were very wide, suggesting low accuracy of the risk estimates [46]. In “The Infant Development and Environment Study” (TIDES), recruiting 705 pregnant women, the average first and third trimester urinary MEP concentration was positively associated with GDM, whereas first-trimester urinary mono-(3-carboxypropyl) phthalate (MCPP) concentration was inversely associated. For mono-n-butylphthalate (MNBP), only the averaged first- and third-trimester urinary concentration was associated with the risk of IGT [47]. In a small cross-sectional study, women with the highest urinary concentrations of mono-iso-butyl phthalate (MIBP) and mono-benzyl phthalate (MBZP) had lower blood glucose levels at the time of GDM diagnosis compared to women with lower urinary concentrations of such phthalates metabolites [48]. These studies show that the effects are not only dependent on the EDCs, but also on the window of exposure.

In conclusion, data are, so far, not sufficient and sometimes contradictory to draw a clear conclusion on EDCs exposure and GDM occurrence, but the field is of interest. The discrepancies in the literature results could be attributed to many factors such as time windows of exposure assessment, the choice of a unique urine sampling to assess exposure, criteria used to diagnose GDM, and lack of information on cocktail exposure.

### 3.2. Role of Social Stress Exposure in Early Life and MD

Stress is defined as a condition that disturbs the physiology of an individual. Stress induces a physiological and behavioral response [49]. It has been shown than childhood trauma can alter brain development as well as immunological and endocrine functions. This physiological response aims to adapt to harsh environments and is part of allostatic load. Allostatic load is defined by the wear and tear on physiological systems in response to stressors; an excessive allostatic load can lead to poor health with age [50].

The association between adverse childhood experiences (ACE) and MD at adulthood was reviewed in 2008 by Thomas et al. [51]. Since then, many cross-sectional and observation epidemiological studies based on various cohorts have shown an association between ACE and MD in later life [52,53]. It has been shown that socioeconomic adversity in early life are important for the development of diseases including MD [54,55,56]. Adverse conditions in early life heighten the risk for adverse conditions in adulthood, which might lead to illness [57]. An association between early life Socio Economic Position (SEP) and the development of T2D from age 30 has been demonstrated in a Danish cohort [58]. Nandi et al. (2012) showed similar evidence for early socioeconomic adversity and diabetes occurrence in adult based on a US longitudinal study [56]. The analysis of a violence questionnaire addressed to a large US longitudinal cohort study of women, the Nurses’ Health Study II (NHSII), associated physical and sexual abuse in children and adolescent and type 2 diabetes occurrence in adult women that is dependent on higher Body Mass Index [59]. The CARDIA (Coronary Artery Risk Development in Young Adult) cohort in the US highlights that low childhood Socio Economic Status and adverse early family environments are associated with high BMI [60].

The question of a role of early life stress as a risk factor for MD in later life has also been studied in minority populations. Indeed, in the US, a large cohort of Black women demonstrated that early life sexual and physical abuse was associated with an increased risk of obesity in adulthood. This association appears dependent on confounding factors such as health behaviors, reproductive history, and mental health, but those confounding factors did not fully account for the associations between early adversity and obesity [61]. Bellis et al. demonstrated an association between cumulative ACE (at least 4) and diabetes in an English cohort [62].

These cross-sectional epidemiological studies highlighted a link between ACE and MD development in later life. Kinetics and understanding of the actors/factors involved in the association between ACE and MD will come from longitudinal epidemiological studies that are still sparse, as well as experimental animal models that are discussed below.

## 4. Evidence for a Role of ELAE on MD Occurrence in Animal Models

### 4.1. Link between Environmental Contaminants and Weight Gain

Recent studies in rodents reported that perinatal (pregnancy and lactation) oral exposure to BPA increased body weight in adult male and female offspring independently of nutritional challenges [63]. Indeed, the effects of BPA exposure during mouse pregnancy were shown to mimic the effects of a high-fat diet in altering glucose and lipid metabolism [64]. In our recent work, young mice perinatally exposed to BPA were leaner and exhibited less perigonadal WAT (gWAT) than vehicle offspring. However, BPA mice recovered rapidly to a normal weight that drifted into an increased body and gWAT weight compared to vehicle offspring [65]. Some in vivo animal studies have not shown effects of BPA on weight gain [66,67].

Discordant results between studies could be attributed to various experimental conditions including animal strain, doses, developmental windows of exposure, and rodents’ housing and diet, which are likely responsible for the discordant results. Of note, studies measuring body fat in addition to body weight are more relevant to address conclusions on MD [68]. In any case, short-term BPA exposure during pregnancy can affect metabolic programming (weight and glucose metabolism) in the offspring later in life.

Prenatal exposure of mice to DEHP reported not only an increased body weight, but also body fat in male offspring; this result has been reproduced by different labs in different animal models [69,70].

### 4.2. Link between Environmental Contaminants and T2D

Glucose intolerance and loss of insulin sensitivity were observed in young and elder mice perinatally exposed to BPA. Similar metabolic disorders have been demonstrated in BPA-exposed animals at an early age with changes in food intake [71]. We could not associate this obese phenotype with pancreatic inflammation, whatever the age of the mice. Interestingly, we provide evidence that perinatal exposure to BPA leads to metabolic disorders including obese phenotype. This phenotype is characterized by inflammatory M1 macrophages infiltration of gWAT in aging offspring, as observed in obese humans [72]. In Malaisé et al.’s (2017) study, the M1 macrophages infiltration of gWAT is associated with IL-17 overexpression [65]. This finding is in accordance with other studies demonstrating tight positive correlation between IL-17 inflammation in WAT and obesity [73,74]. Mice perinatally exposed to BPA displayed immune homeostasis disturbances in association with gut microbiota dysbiosis in early adulthood and altered glucose tolerance, then while aging they developed obese phenotype.

Similarly, rats exposed to DEHP throughout gestation and perinatal development exhibited hyperglycemia in the presence of reduced insulin levels along with reductions in β-cell mass, reduced islet insulin content, and disruptions in β-cell ultrastructure [75]. Hyperglycemia with reduced insulin levels was also found in females exposed to diethylhexyl phthalate throughout the gestation/perinatal period [76].

Collectively, these data suggest that impairments in insulin action may result from exposure to a variety of environmental EDCs; however, dose, duration, and models may alter the phenotypic response to these compounds. This suggests that exposure during pregnancy may alter the long-term metabolic trajectory of both the mother and her offspring.

### 4.3. Link between Social Stress and Metabolic Disorders

Neonatal maternal separation (MS) is a stress model widely used in rodents as a paradigm of early life adverse events that can occur in humans. Scattering evidences suggesting consequences of MS on metabolism have already been published and detailed hereinafter. MS alone did not affect glucose intolerance in Sprague–Dawley male rats aged 8 months, but diminished the expression of insulin receptor in muscle and the concentration of IGF-1 in serum [77]. However, MS combined with post-weaning social isolation in Sprague–Dawley rats induced glucose intolerance at Post Natal Day (PND) 180, associated with higher corticosterone concentrations [78] reinforcing the hypothesis of higher vulnerability to additional risk factors and an increase in allostatic load [77]. Finally, shorter MS protocol increased body weight, glucose, and insulin response after arginine-stimulation in male Sprague–Dawley rats aged between PND105 and PND133 [79].

Finally, we demonstrated in a mouse model that MS induced MD without nutritional challenge in PND350 mice. MD are aging disorders and we hypothesized that MS consequences on metabolism will appear in aged animals. This MD was characterized by impaired glucose tolerance and loss of insulin sensitivity without modification of BMI [80].

## 5. Hypothesis of Different Mechanisms Leading to MD

Interestingly, obesity and T2D are associated with intestinal barrier impairment and systemic inflammation. Indeed, human cohort studies reported that obesity and/or type 2 diabetes (T2D) have been associated with increased intestinal permeability [81,82], IgG response against specific *Escherichia coli* (*E. coli*) [83], dysbiosis [84], and low grade systemic inflammation [85]. Common mouse models of obesity associated with hyperglycemia are genetic models (ob/ob and db/db, respectively, deficient for leptin and leptin receptor) and induced diet models such as high-fat or Western diets. In those mouse models, as in humans patients, a defect of the intestinal barrier and low-grade inflammation are observed, even before the onset of obesity and hyperglycemia [86].

Based on our work on the long-term consequences of early life exposure to food contaminants and social stress on the intestinal barrier and systemic inflammation, we questioned their role in MD occurrence.

Moreover, epigenetics have been identified as an important contributor to the vulnerability of the developmental period. Indeed, epigenetic signaling regulates gene expression, which controls development. Epigenetic changes provide biochemical evidence of the deleterious effects of adverse conditions during development and subsequent diseases, including metabolic diseases [87].

### 5.1. Actors Involved in MD

With aging, low-grade inflammation and intestinal permeability are increasing in mice [88] and humans [89]. This partly contributes to the higher susceptibility to chronic metabolic diseases in aging populations and suggests a potential role of intestinal barrier defects and low-grade inflammation in triggering and/or maintaining metabolic disorders.

#### 5.1.1. Intestinal Barrier—Contribution in Immune Response Commitment

The intestinal barrier is made of different actors aiming to defend the host toward potentially harmful luminal components (pathogens, toxins etc.). The intestinal barrier is composed of motility, microbiota, intestinal epithelium (defining intestinal permeability), and gut-associated immune system.

MD is associated with gut microbiota dysbiosis. In patients with T2D, microbiota dysbiosis was mainly characterized by a moderate degree of enrichment in Bacteroidetes and Betaproteobacteria, associated with a decrease in Clostridia within Firmicutes phylum [90,91,92]. A metagenome-wide association study in T2D patients showed that microbiota dysbiosis was mainly due to an increase in pathobionts such as *Bacteroides vulgatus* and *Enterococcus faecalis* as well as Enterobacteriaceae such as *Proteus mirabilis* and *E. coli* [91,93]. *B. vulgatus* and *E. coli* are suspected of driving insulin resistance [91,92].

Intestinal hyperpermeability was observed in obese [94] but not in T2D patients [95]. The only available study on intestinal permeability in T2D patients was performed in a small cohort of 18 patients. Nevertheless, an increase in both para- and trans-cellular intestinal permeability was observed in mouse models of obesity [81,82].

MD is associated with intestinal barrier impairment. The intestinal barrier prevents inappropriate immune response toward luminal content. In MD, this intestinal barrier defect is associated with systemic immune response toward luminal content that is detailed in the following section.

#### 5.1.2. Immune System—Chronic low-Grade Inflammation

The link between inflammation and MD has been reviewed by Hotamisligil (2006) [96]. Abnormal humoral immune response to commensal microbiota has been observed in obese diabetic patients and mouse models [83]. A decrease in Treg and an increase in Th1 and Th17 cells were observed in the blood of T2D patients [97]. Furthermore, in both genetic and diet-induced obese/diabetic mice, a waning of Treg was observed in adipose tissue [88]. Furthermore, it has been shown that there is a defect of B and T cells populations in the blood of patients suffering from T2D [97,98]. T2D patients had lower Treg and higher Th1 and Th17 populations associated with elevated pro-inflammatory cytokines concentration in circulating blood [97]. Not only T cells but also B cells are impaired in T2D patients and contribute to overactivation of T cells [98].

Inflammation associated with an increase in plasmatic anti-microbiota IgG titers and intestinal permeability are common features of metabolic disorders, including liver steatosis [99,100], obesity [81,82], and diabetes [83].

Besides epidemiological data, experimental studies reveal a relationship between immune signature and MD. Indeed, HFD-induced obesity and T2D is associated with lower intestinal Th17 and Th22 immune cell populations [101,102,103]. Furthermore, RORγt KO mice that are deficient for Th17/22 cells exhibited a mild glucose intolerance [101]. Metabolic disorders induced by HFD (High Fat Diet) could be dampened by increasing IL-22 [103,104]. Higher IL-22 concentration not only moderates MD but also restores microbiota [104,105] and the intestinal barrier [105].

It has been demonstrated that low birth weight individuals have higher concentrations of inflammatory markers in adulthood that might contribute to the development of MD [106].

So far, no consensus exists among scientists to decipher if low-grade inflammation is a cause or a consequence of metabolic disorders. This is a vicious circle since obesity drives production of inflammatory markers by white adipose tissue and contributes to low-grade inflammation [107] and endotoxemia (leakage of endotoxin in serum due to higher intestinal permeability) which can induce obesity and insulin resistance [108]. Altogether, these studies showed that the intestinal barrier and immune response in gastrointestinal tract and at systemic level are affected in MD.

### 5.2. Role of Intestinal Barrier (IB) and Inflammation in Early Life Exposure to EDC-Induced MD

As MD is characterized by IB impairment, associated with systemic inflammatory immune response, we wonder if food contaminants (BPA and phthalates) and social stress also present those features. If so, longitudinal experimental models will be of great interest to decipher if IB defects and inflammation are causes or consequences of MD. This information would be crucial to identify preventing strategies for MD.

We previously demonstrated that perinatal exposure to BPA in offspring mice leads to obesogenic effects, with close characteristics to human obese phenotype. Interestingly, this obesogenic phenotype is associated with glucose intolerance and higher splenic Th1 and Th17 responses, attesting to systemic inflammation [65]. These results are in accordance with Luo et al.’s (2016) study demonstrating similar Th1/Th17 immune deviation in the spleen of male and female mouse offspring perinatally exposed to BPA [109].

BPA perinatal exposure in rats has been described to deeply affect homeostasis of the gut immune system [110] and represents a risk factor to developing pro-inflammatory conditions in adulthood [111]. Perinatal exposure to BPA in our mouse model leads to a loss of Th1/Th17 cell subsets from *Lamina Propria*. Interestingly, recent publications reported similar observations in mice fed with HFD. Indeed, Garidou et al. showed that ileum microbiota dysbiosis induced by HFD trigger a decrease in Th17 cells of the *Lamina Propria*. [101]. The decrease in Th17 and Th1 responses in the small intestine induced by BPA in our mouse model is associated with intestinal dysbiosis. Indeed, BPA perinatal exposure in mice altered gut microbiota development in offspring, driven by a decrease in *bifidobacteria* known to display anti-inflammatory properties [112]. Microbial changes, i.e., a decrease in *bifidobacteria* or/and impairment of intestinal immune homeostasis in young adult mice (i.e., PND45), could drive altered microbial pattern observed in aging mice. The altered gut microbiota following perinatal BPA treatment could also result from a defect of intestinal defenses. A loss of IgA and anti-microbial activity (lysozyme) was observed in the feces of adult mice [65,113]. These animals also have a lower expression of polymeric IgR (pIgR) (receptor involved in basolateral to apical transport of IgA), which could contribute to lower IgA concentrations in feces.

The role of BPA in the occurrence of metabolic disorders remains unclear. However, a consensus exists identifying perinatal period as a critical window for BPA exposure, with evidence of imprinting on immune system development and function for life [65,71,110,114].

However, further studies are necessary to confirm or refute the link between EDCs exposure and T2D. It is also mandatory to include data on the duration and intensity of EDCs exposure, and to recognize early-induced molecular imprinting (epigenetic modifications). Another key issue is that EDCs can exhibit health-threatening effects at low doses (below those used in traditional toxicological studies) that may not be predicted by effects at higher doses, especially in the case of BPA. In the meantime, medical practitioners, especially when dealing with the most sensitive periods of life, such as during infancy and pregnancy, should adopt a precautionary principle towards EDCs exposure.

### 5.3. Role of Intestinal Barrier and Inflammation in Early Life Stress-Induced MD

Stress has many consequences on the endocrine system that could trigger MD. Indeed, stress may lead to the elevation of glucocorticoids and thereby stimulate insulin release and appetite, leading to an elevated desire to eat and altered eating patterns. Positive neuroendocrine feedback associated with the intake of certain foods may reduce feelings of stress and ultimately reinforce unhealthy eating patterns [115]. Stress is associated with hypothalamic-pituitary-adrenal dysregulation, increased blood levels of insulin, insulin resistance, and visceral fat accumulation [115,116]. Finally, metabolic and hormonal disruption due to stress may impact adipose tissues and influence insulin levels and the risk for insulin resistance [117,118]. In this overview, we aim to propose a new cascade of events that might also contribute to MD occurrence, i.e., intestinal barrier defects and inflammation.

Neonatal maternal separation (MS) is a stress model widely used in rodents as a paradigm of early life adverse events that can occur in humans but is also widely used as a model of Irritable Bowel Syndrome (IBS) (see [119]).

The occurrence of early stressful events is considered as a contributing factor triggering and/or maintaining IBS, a functional gastrointestinal disorder [120,121]. Beside visceral pain, IBS is also characterized by increased intestinal permeability [122], microbiota dysbiosis (for review [123]), and an increased state of activation of immune cells [124], although this last observation is still under debate. Early life events draw particular attention since they are associated with IBS susceptibility [125,126,127]. This association between early life stress and gastrointestinal disorders is of particular interest and questioned their role in MD occurrence in later life. This relationship between IBS and MD was highlighted by epidemiological studies. First, they show that both stress and IBS are positively correlated with higher HbA1c (glycated hemoglobin) in patients suffering from type 2 diabetes [128]. Then, they establish a link between IBS and the development of metabolic disorders independently of dietary patterns [129,130]. These data are key to associate early life stressful events, gastrointestinal outcomes, and MD. We then wondered if we could find such an association between early life stressful events, inflammation, and MD.

Childhood victimization is positively correlated with higher plasmatic CRP concentration in young human adult [131]. Immune response is dependent of socioeconomic status and associated with many pathologies, including MD [132,133]. Inflammation represents a first response to social stress and by the time of this immune response, no MD are observed, suggesting that inflammation precedes MD. Indeed, Danese et al. reported elevated concentrations of C-Reactive Protein (CRP) in children who were exposed to physical abuse and experience depression. This observation was independent of confounders such as family, socio-economic circumstances, obesity, or body temperature [134]. The same group performed a prospective study on the New Zealand Dunedin Multidisciplinary Health and Development cohort to measure inflammatory markers at adulthood and a potential association with childhood maltreatment. They showed that stress-induced inflammation in childhood carries on into adulthood. Children exposed to maternal rejection, harsh parenting, disruptive caregiver changes, and physical or sexual abuses were more likely to have higher levels of inflammatory markers in adulthood that non maltreated children [135]. In the same cohort, adverse childhood experiences (ACE) were associated with MD in adulthood [136]. These cross-sectional and longitudinal studies show that first ACE induce inflammation (first response) and then, secondarily, MD, and give some element to define the causes and consequences on inflammation and MD.

Based on the European Prospective Investigation into Cancer and Nutrition (EPIC) cohort, it has been shown that early social inequity affects physiology at adulthood and induces higher inflammatory status [133]. Low childhood socioeconomic conditions are associated with chronic low-grade inflammation during adolescence [137]. Based on six European cohort studies within the lifepad project including more than 17,000 participants, it has been demonstrated that CRP is the pro-inflammatory marker with a stronger association with defect of education [138]. Still based on the lifepad consortium, it has been demonstrated that early socioeconomic disadvantages were strongly associated with higher levels of CRP and demonstrate biological consequences of ACE [139].

It is important to note that inflammatory markers observed in children are dependent of the ACE (maternal mental health problems and physical abuse or parental conflict and emotional abuse) and can affect differently girls and boys [140]. In a UK Household Longitudinal Study, it was demonstrated that the inflammatory system and, to a lesser extent, the metabolic system consistently drove (across age groups and gender) the observed values of the Biological Health Score independently of social status and exposure [141]. They paved the way to show that social environments modify biological functions that are involved in health inequalities at later ages.

Furthermore, psychological stress affects maternal gastrointestinal (GI) permeability, leading to low-grade inflammation which negatively impacts fetal development by inducing neonatal stress [142]. Data from experimental animal studies also reinforce the hypothesis for a role of stress-induced IB defects and inflammation in MD development. MS increased circulating IL-6 in PND15 rats [143]. The same group demonstrated that MS was associated with an increased ratio CD4/CD8 in spleens of PND21 [144].

In adult rats (PND80) submitted to MS, a defect of colonic permeability was associated with a local (MPO, *Ifn**γ* and *Il-1**β*) and systemic inflammation of the liver and spleen (IFNγ and IL-1β) [145]. Increased intestinal transcellular permeability in small intestine and microbiota dysbiosis were associated with local (increased IgA and decreased ILC3) and systemic inflammation (IFNγ and IgG directed against microbiota) in a mice model of MS aged of PND50 [146]. In a longitudinal experimental study in rats, MS induced bacterial penetration in colons at PND20, 25, and 30 associated with systemic (spleen) penetration at PND20. Colonic hyperpermeability was observed at PND20 and PND25 but not PND30, and increased colonic MPO at PND20 [147]. These results show that defects of the intestinal barrier and inflammation are primary responses to MS that can fade with time. In none of those studies, the MS effect on inflammation (local and systemic) and intestinal barrier impairment was associated with metabolic disorders.

Based on our longitudinal studies on MS models in mice, we could demonstrate that intestinal barrier defects and inflammation precede MD. Indeed, we demonstrated that MS triggers an IBS-like syndrome in young mice associated with IB impairment and systemic inflammation, but no MD [146,148]. Then, if we age those mice submitted to MS that develop IB impairment and inflammation in young adulthood, they develop T2D features in later life [80]. This experimental model is of great interest to determine that IB defect and inflammation precede MD development and offer a perspective for prevention strategies.

These results are of particular interest and highlight all stressors of IB and inflammation as potential risk factors for MD development in later life.

### 5.4. Role of Epigenetics in Early Imprinting

MD is associated with aged-related conditions such as pro-inflammatory states and IB defects, as described below. In this framework, aging is modelled as a progressive decline of the integrity and/or level of functioning across multiple organ systems [149]. Then, MD are often conceptualized as a kind of premature or accelerated aging with a potential role of epigenetics. Vaseirman and al. (2019) reviewed evidence of the role of developmental epigenetic variation in the pathogenesis of T2D [150].

Transgenerational inheritance of metabolic diseases could be attributed to stressors, including high-fat, high-sugar diets, low protein diets, and EDCs. Many recent papers showed that EDCs exposure only in pregnant F0 animals will have deleterious outcomes, at least until the F3 generation [151]. This is striking, as F0 and F1 animals are directly exposed to EDCs, whereas the F2 generation is exposed as germ cells within the gestating F1 animals. The F3 generation is the first generation that did not encounter the EDCs; therefore, effects observed in F3 and beyond are considered to be transgenerational and permanent, and different from multigenerational effects observed in F1 and F2 animals [152,153,154].

Many of the EDCs’ actions involving nuclear receptors are tightly linked to epigenetic changes known to be important in transgenerational effects. Chemical exposure or diet modification are known to alter DNA methylation, histones, and copy number variants that contribute to transgenerational phenotype.

BPA and phthalates were indicated as sources of epigenetic disruption [155]. Epigenetics are defined by alterations in gene expression and not in DNA sequences that lead to heritable phenotype. These modifications are usually due to a lower gene expression. The fetal state is particularly vulnerable to epigenetic insults due to a high DNA synthesis rate and because of the establishment of the complex machinery modulating DNA methylation and chromatin organization [156]. Studies investigating EDCs’ impact on fetal–placental epigenetics in human pregnancy are sparse because of the ethical issues associated to the collection of samples during the first trimester of pregnancy. EDCs induce genome alterations in pregnancy or early life and enchain in a decreased expression of pancreatic/duodenal homeobox 1 transcription factor gene (PDX-1) associated with an increase in T2D (review in [157]), suggesting that in utero exposure to impaired nutrition is a risk for obesity and diabetes progression in adulthood. EDCs confine the stock of essential metabolic substrates to the fetus and cause intrauterine growth retardation, presenting as fetal starvation and the metabolic basis that triggers diabetes progression in PDX-1.

The BPA substitute BPS was reported to alter the placental expression of P-glycoprotein (P-gp), one of the main efflux transporters to xenobiotics encoded by the ABCB1 gene. Speidel and colleagues demonstrated that acute exposure to BPS (0.5 nM) induced a significant haplotype-dependent decrease in ABCB1 promoter activity in CRL-1584 human trophoblast cell lines. However, chronic BPS exposure (0.3 nM) induced a significant haplotype-dependent promoter activity increase, impacting P-gp levels and fetal exposure to xenobiotics coming from the maternal circulation [158].

Few studies of transgenerational inheritance have paid attention to the effects of EDCs exposure on the immune system. We observed that early life exposure to BPS strongly impaired intestinal immune status at multigenerational (F1 and F2 exposure) or transgenerational patterns (F3 exposure). Both F1 and F2 male offspring developed inflammatory responses in ileum and colon in adulthood after F0 mothers were exposed to BPS [159]. The response is sex and generation-dependent as a decrease in inflammatory response was observed in the F1 female generation and the opposite in the F3 generation, i.e., inflammation. Bansal et al. (2017) also observed that maternal (F0) exposure to BPA has multigenerational sex-dependent outcomes; first (F1) and second generation (F2) adult female offspring were unaffected, but F1 and F2 adult male offspring had an increased percentage of body fat and reduced glucose stimulated insulin secretion [160].

In our model of F0 exposed to BPS, the persistence of the metabolic abnormalities in the third generation is attributed to epigenetic modifications. The exposure of germ cells is important to take into account when addressing multigenerational effects. However, the ability to induce a permanent epigenetic alteration in the germ cells, without follow-up chronic exposure, suggests a novel form of inheritance, which could have a much greater impact on biology, disease etiology, and evolution. A wide variety of environmental factors, from nutrition to toxicants such as BPA or BPS, have now been shown to promote the epigenetic transgenerational inheritance of disease or phenotypic variation [69,159,161]. Important transgenerational effects of BPS in male offspring can be explained by the decrease in intestinal inflammation observed only in F3 offspring [159]. Moreover, epigenetic control of intestinal barrier functions and inflammation has been recently described [162].

Thus, BPS, as a so-called safer alternative to BPA, does not appear so safe, as it has detrimental effects on fetal programming and transgenerational deleterious outcomes. These data open new perspectives into the understanding of EDCs’ influence on the prenatal development of adult health susceptibilities and pregnancy-related disorders.

In mice, early life adverse events (ELAE) can induce DNA methylation of glucocorticoid receptor [163]. Studies have also demonstrated that early life stress and deleterious relationships with mothers could induce epigenetic signatures via DNA methylation [164]. Interestingly, in monkeys, the quality of infant-mother relationships has consequences on the methylation of genes involved in immune response [165].

DNA methylation-based measures of the difference between epigenetic age and chronological age were demonstrated to be predictors of mortality risk in meta-analysis of four independent cohorts. The mortality risk appears to be a combination of different factors including cardiovascular disease, high blood pressure, and diabetes [166]. Based on an EnviroGenoMarkers (EGM) study, regrouping participants from an Italian EPIC cohort, less-advantaged SEP participants exhibit, later in life, a lower inflammatory methylome score in peripheral blood mononuclear cells, suggesting an overall increased expression of the corresponding inflammatory genes or proteins. These data suggest that epigenetics could contribute to stress-induced inflammation and contribute to later development of MD [167]. Finally, it has been demonstrated that MS-induced visceral hypersensitivity (IB defect) is transferred across generations and that this effect likely depends upon maternal care [168].

To conclude, it is of great interest to have in mind that disadvantaged social conditions should be considered as an accelerator to aging processes. Early social adversity has biological imprinting. Social-to-biological processes that are induced in early life are likely to lead to socially differentiated biological states and health inequalities.

## 6. Conclusions and Perspective on Enriched Environments

In summary, we combined studies highlighting a role of ELAE focusing on BPA/phthalates and social stress on MD development. This overview synthetizes epidemiological studies and experimental data on animals to support the hypothesis that MD are not only related to our way of eating and moving but also to early origins, and can find their roots in DOHaD.

Within this overview and based on our expertise on IB and immune response, we hypothesize that the impairment of IB and inflammation induced by ELAE are risk factors for MD. Indeed, MD have been associated with IB defects and inflammation. We quote studies supporting the hypothesis that IB defects and inflammation precedes ELAE-induced MD. Exacerbated immune response driving inflammation is closely linked to metabolic diseases and obesity developments [72,169]. The close association between immune and metabolic status led to an emerging field of research named “immunometabolism” [170]. The ability of pro-inflammatory cytokines to impair cellular insulin signaling clearly identify inflammation as a key feature in MD [171]. Special emphasis has been placed on the intestinal immune system. Indeed, increased intestinal permeability in MD could modify gut associated immune system and drive and/or maintain MD. Intestinal CD4^+^ T cells are committed to various subsets, notably inflammatory (Th17/Th1) and regulatory cells (Treg), with Th17 cells being the most abundant CD4^+^ T cells in mucosal tissues [172,173,174]. They secrete isoforms of IL-17 and/or IL-22, known to protect the host from fungi, parasites, and pathogenic bacteria. The gut microbiota and the shape of IB also play a crucial role in the development of obesity and diabetes, and its relationship with the immune system makes it a great point of interest. However, further data form longitudinal studies in epidemiology and experimental animal studies are needed to decipher a causal role of IB defects and inflammation in ELAE-induced MD. However, if this hypothesis is verified, it will provide new therapeutic strategies to reduce the epidemic of MD.

The positive effects of environment stimulation and enrichment have been studied for many years. The concept of “enriched environment” was proposed in the 1940s by Hebb [175]. If prevention strategies failed, it would be interesting to identify therapeutic strategies to avoid or at least dampen the negative effects of ELAE on the development of MD at adulthood. Robust effects of environmental enrichment in animals suggest that a similar species-typical enrichment of humans could diminish the adverse consequences of urbanization (stress, Western diet, lack of physical activity). The relationship between our living environment and human health appears tightly connected. Indeed, lifestyle and environment are not only social issues, but also biological necessities for public health. In Europe, many programs aim to promote positive and enriched environments to counterbalance to detrimental outcome of early life exposure to toxins and stress inherent with our modern ways of life (Women Engage for a Common Future https://www.wecf.org/, accessed on 20 February 2021). Experimental data coming from pre-clinical studies support the role of an enriched environment in positive health outcomes, and encouraging research supporting these practices in DOHaD studies would be of great interest [176].

## Data Availability

Not applicable.

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
