# Peer review of "Early Life Exposure to Food Contaminants and Social Stress as Risk Factor for Metabolic Disorders Occurrence?—An Overview"

_biomolecules, 2021, doi:10.3390/biom11050687_

Round 1
Reviewer 1 Report
This is a review article regarding evidence for early life exposure to food contaminant and social stress as risk factor for metabolic disorders (MD) occurrence. Overall, the manuscript was written well, but there are some comments.
- The authors are trying to build up the hypothesis on the causal role of food contaminant and social stress in the DOHaD hypothesis. However, since the authors presented conflicting results and no direct evidence to support their hypothesis, the conclusion seemed too optimistic as a review. This paper should be categorized as “perspectives” rather than review.
- Food contaminants and social stress are completely different factors. Mixture of these two factors is confusing and misleading. These two factors should be discussed separately.
- The authors propose the several key terms such as DOHaD, food contaminant, social stress, intestinal barrier disruption etc., but associations between the terms were not clearly presented but rather speculative. This suggests that each term may be related to other terms in some way, but the significance may be small.
- Including graphical abstract may be better for readers to understand the contents of this paper.
Author Response
- The authors are trying to build up the hypothesis on the causal role of food contaminant and social stress in the DOHaD hypothesis. However, since the authors presented conflicting results and no direct evidence to support their hypothesis, the conclusion seemed too optimistic as a review. This paper should be categorized as “perspectives” rather than review.
We thank the reviewer for this interesting remark. Indeed, based on a review of literature, we aim to present a new hypothesis highlighting early life exposure to food contaminant and social stress as risks factors for metabolic disorders occurrence. In order to achieve this goal in an objective way, we quoted and discussed data supporting or not this hypothesis. Even though, data could appear conflicting something, the majority of literature make our hypothesis reasonable and worthwhile to address. However, as correctly noticed by the reviewer this is an hypothesis/point of view. We modify the title of the manuscript “Evidence for early life exposure to food contaminant and social stress as risk factor for metabolic disorders occurrence?: an overview.” In order to clarify the position of the manuscript. Furthermore, review has been replaced by overview in the all manuscript. We hope those modifications will fulfill reviewer request.
- Food contaminants and social stress are completely different factors. Mixture of these two factors is confusing and misleading. These two factors should be discussed separately.
We do agree with the reviewer that food contaminants and social stress are completely different factors. However, this is quite interesting to highlight that based on an overview of literature, they both appear as risk factors for metabolic disorders when encounter in early life. Even though, they are indeed different factors they seem to involve the same mechanisms intestinal barrier disruption, inflammation, epigenetic modifications and MD as pathological outcome. We think that discussing both models in parallel strengthen our hypothesis. We apologize that it appear confusing for the reviewer. In order to clarify this issue and avoid any misleading of the reader, we had better explain the choice of those models and their complementarity.
The following sentences have been added to the manuscript:
Abstract:
“Indeed, summing up data on both ELAE models in parallel allowed us to identify common patterns that appear worthwhile to study in MD etiology.”
Section 2. early origins of metabolic disorders:
“Based on those studies showing that nutritional restriction or excess in early life are associated with outcomes on offspring metabolism, both authors took advantage of their expertise in the field of long time consequences of early life stressful adverse events and exposure to food contaminants to propose that they could be consider as cofounding risks factors in metabolic disorders development.
We developed this hypothesis based on epidemiological and experimental studies that allow working on control models with limited confusing factors and identifying if perinatal exposure to food contaminants or stressful events beside their association with food are risk factors for development of MD at adulthood. Developing this hypothesis, from two different models of ELAE, appears like a strength to identify a plausible link between ELAE and MD occurrence and decipher the mechanisms involved.”
- The authors propose the several key terms such as DOHaD, food contaminant, social stress, intestinal barrier disruption etc., but associations between the terms were not clearly presented but rather speculative. This suggests that each term may be related to other terms in some way, but the significance may be small.
We agree with the reviewer that we only present association between food contaminant and social stress in early life (dohad concept) and development of MD. Those associations are based on epidemiological and experimental studies and compile results lead tho the reasonable hypothesis that early life adverse event could contribute to MD but so far there is no evidence of a causative role. We added the following statement at the end of the introduction section. We hope it will prevent any confusing or misleading aim of the manuscript.
“The aim of this overview is to compile evidences highlighting a relationship between ELAE-induced (food contaminants or stress) intestinal barrier disruption, inflammation, epigenetics and occurrence of MD. Even though, there are no striking studies linking ELAE and MD. Our goal is to combine evidences based on a review of the literature and offer a new field of research and perspectives on a role of DOHaD in MD occurrence.”
- Including graphical abstract may be better for readers to understand the contents of this paper.
We thank the reviewer for this excellent remark. We provided a graphical abstract in the revised version of the manuscript.
We hope this graphical abstract will indeed provide a quick and comprehensible sump up of the manuscript content.

Reviewer 2 Report
Here are my suggested edits:
Abstract
Sentence 1: The global prevalence of obesity has been increasing these last years and is now the major public health challenge worldwide.8iiiii
Sentence 3: Based on human epidemiological and pre-clinical experimental studies, this review questioned the role of non-nutritional components as contributors on the epidemic of MD with a special emphasize on food contaminant and social stress.
Last Sentence: This review sums up evidences that MD could have developmental origins and that ELAE are risk factors for MD at adulthood independently of nutritional status.
Introduction
Sentence 1: Over the last decades, metabolic disorders (MD) encompassing obesity, ….
Sentence 3: Most of the world's population live in countries where overweight and obesity kill more people……….
Paragraph 2—Sentence 2-They are tightly linked to way of life specially food intake and sedentary lifestyle.
Page 2—line 1---carefully addressed as highlighted by the mutual report from WHO, UNICEF and Lancet commissions [1].
Paragraph 2---line 1--4The developmental origin of health and diseases (DOHaD)….. P
- Early origins of metabolic disorders
Line 5 ….that famine during early gestation led to heavier babies at risk to develop obesity…
In the Nigerian famine, occurring during civil war (1967-1970), fetus exposure to famine was also associated with increased risk of MD [7]. Data from famines were of great interest to show the association between ELAE and risk to develop MD. However, famines are not only characterized by food restriction, but also by the stressful environment that appeared as confounding factors
2nd Paragraph
At the other hand, obesity due to excess nutrients and/or feeding with a western diet and lack of physical activity drive women with high Body Mass Index (BMI) to enter the first trimester with MD and/or experience significant gestational weight gain. The American College of Obstetricians and Gynecologists identified maternal obesity as the greatest public health risk in pregnancy and the higher number of large gestational age infants are born to mothers with obesity [8]. The intrauterine environment in obesity is characterized by nutrient excess and associated with increased risk for childhood obesity [9].
Furthermore, greater weight gain in the first trimester was linked with risk of overweight in children [14].
However, high BMI in mothers is usually associated with diet, and, besides nutrients, diet, especially highly transformed diet, also contained food contaminants. Furthermore, epidemiological evidence supports, not only the need for early-life interventions to reduce the obesity and diabetes burden in later life, but also preconception intervention (for review [15]) highlighting a potential role of epigenetic that will be discussed in this review.
Paragraph 3: The role of early life exposure to food contaminant in MD development will be discussed in this review.
Paragraph 4---Interestingly, the DOHaD concept was initiated by a study demonstrating an association between low birth weight and risk of ischaemic heart disease in later life [16]. It was clearly demonstrated since the 90s that early life exposure can imprint the metabolism of the offspring especially that maternal and postnatal nutrition are risk factor for development of T2D [17]. This field of research is of great interest, as a better understanding in behavior affecting MD will provide prevention strategies and promotion of public health regarding metabolic disorders. Many cohorts in children (TEDDY, HAPO, ABCD, TODAY, PEACHES, EPOCH) addressed the role of early life exposures in T2D occurrence (for review [18]). Most of those cohorts focused on how maternal metabolism (including gestational diabetes mellitus, GDM) and their nutritional status will affect offspring metabolism.
Last sentence--We developed this hypothesis based on epidemiological and experimental studies that allow working on control models with limited confusing factors and identifying if perinatal exposure to food contaminants or stressful events beside their association with food are risk factors for development of MD at adulthood.
3.1. Role of environmental contaminants
……and as such interfere with organ development, reproduction, immunity, metabolism….
3rd paragraph—2nd sentence---This is facilitated by the ability of EDCs to cross placental barrier and then, to enter in the fetal circulation [21].
4th paragraph---3rd sentence--- First, predisposition to obesity starts as early as in the fetal condition….
3.1.1. Focus on endocrine disruptors chemicals (EDC): bisphenols and phthalates
4th paragraph---line 9----Special attention to confounding factors is needed; indeed diet is not the only source of BPA exposure.
Last sentence---While those data are insufficient to draw clear-cut conclusions, they support the hypothesis that…….
3.1.2. EDCs could trigger MD in offspring by inducing gestational diabetes mellitus (GDM)
Line 5----- Complications of GDM for the offspring are increased birth weight and predisposition for MD (American diabetes association 2014).
Line 8---Based on various cohorts, many studies investigated a possible association between pregnancy exposure to EDCs and occurrence of GDM; however the results are not clearcut.
Paragraph 2—Line 9--The association between MEP and gestational weight gain showed a U-shape whereas DEHP and gestational weight gain was described by an inverse U-shape [41].
Those studies show that the effects is not only dependent on the EDCs, but also on the window of exposure.
Last paragraph---In conclusion, data are not sufficient and sometimes contradictory to draw a clear conclusion on EDCs exposure and GDM occurrence.
3.2. Role of social stress exposure in early life and MD
1st paragraph---Line 5----Allostatic load is defined by the wear and tear on physiological systems in response to stressors; an excessive allostatic load can lead to poor health with age [49].
2nd paragraph—line 13----- The analysis of a violence questionnaire addressed to a large US longitudinal cohort study of women, the Nurses' Health Study II (NHSII) associated physical and sexual abuse in children and adolescent and type 2 diabetes occurrence in adult women that is dependent on higher Body Mass Index [58].
Last sentence----The CARDIA (Coronary Artery Risk Development in Young Adult) cohort in US highlights that low childhood Socio Economic Status and adverse early family environment are associated with high BMI [59].
- Evidence for a role of ELAE on MD occurrence in animal models
4.1. Link between environmental contaminants and weight gain
Paragraph 2—line 1---Discordant results between studies could be attributed to various experimental conditions including animal strain, doses, developmental windows of exposure, rodents housing and diet which are likely responsible for the discordant results.
Last sentence---Prenatal exposure of mice to DEHP reported not only an increased body weight, but also body fat in male offspring; this result has been reproduced by different labs in different animal models [68,69].
4.2. Link between environmental contaminants and T2D
2nd sentence---Similar metabolic disorders have been demonstrated in BPA-exposed animals at an early age with changes in food intake [70].
4.3. Link between social stress and metabolic disorders
2nd paragraph---line 3---This MD was characterized by impaired glucose tolerance, loss of insulin sensitive without modification of BMI [56].
5.1. Actors involved in MD
Line 1--With aging, low-grade inflammation and intestinal permeability are increasing in mice [83] and human [84].
Line 2---This partly contributes the higher susceptibly to chronic metabolic diseases in aging population and suggest a potential role of intestinal barrier defect and low-grade inflammation in triggering and/or maintaining metabolic disorders.
5.1.1. Intestinal barrier – contribution in immune response commitment
Line 1---Intestinal barrier is made of different actors aiming to defend the host ……
2nd paragraph---line 4---A metagenome-wide association study in T2D patients showed that microbiota dysbiosis was mainly due to increase of pathobionts ……. B. vulgatus and E.coli are suspected of driving insulin resistance [86,87].
4th paragraph—2nd line --- In MD, this intestinal barrier defects is associated with systemic immune response toward luminal content that is detailed in the following section.
5.1.2. Immune system – chronic low grade inflammation
3rd paragraph—line 1---Besides epidemiological data, experimental studies reveal a relationship between immune signature and MD.
So far, no consensus exists among scientists to decipher if low-grade inflammation is a cause or consequence of metabolic disorders. This is a vicious circle since obesity drives production …...
5.3. Role of Intestinal Barrier and inflammation in early life stress-induced MD
4th paragraph—line 9---The same group performed a prospective study on the new-Zeeland Dunedin ……..
4th paragraph---Lines 13-14---- They showed that stress-induced inflammation in child carry on at adulthood. Children exposed to maternal rejection, harsh parenting, disruptive caregiver changes, physical abuse or sexual abuse were more likely to have higher levels of inflammatory markers in adulthood than non maltreated children [130].
In adult rats (PND80) submitted to MS, a defect of colonic permeability was associated with a local (MPO, Ifng and Il-1b) and systemic inflammation liver and spleen (IFNg and IL-1β) [140].
Paragraph 8--In a longitudinal experimental study in rats, MS induced bacterial penetration in colon at PND20, 25 and 30 associated with systemic (spleen) penetration at PND20. Colonic hyperpermeability was observed at PND20 and PND25, but not PND30 and increased colonic MPO at PND20 [142].
Based on our longitudinal studies on MS model in mice, we could demonstrate that intestinal barrier defect and inflammation precede MD. Indeed, we demonstrated that MS trigger IBS like syndrome in young mice associated with IB impairment and systemic inflammation, but not MD [141,143].
5.4. Role of epigenetic in early imprinting
Line 2---. In this framework, aging is modelled as a progressive decline of the integrity and/or level of functioning across multiple organ systems [145]. Then, MDs are often conceptualized as a kind of premature or accelerated aging with a potential role of epigenetic.
Many recent papers showed that EDC exposure only in pregnant F0 animals will have deleterious outcomes at least until the F3 generation [147]. This is striking as F0 and F1 animals are directly exposed to EDCs, whereas the F2 generation is exposed as germ cells within the gestating F1 animals.
Many of the EDCs action involving nuclear receptors tightly linked to epigenetic changes are known to be important in transgenerational effects.
Chemical exposure or diet modification are known to alter DNA methylation, histones and copy number variants that contribute to transgenerational phenotype.
Studies investigating EDCs’ impact on fetal–placental epigenetics in human pregnancy are sparse because of the ethical issues associated to collection of samples during the first trimester of pregnancy. EDCs induce genome alterations in pregnancy or early life and enchain in a decreased expression of pancreatic/duodenal homeobox 1 transcription factor gene (PDX-1) associated with increase of T2D (review in [153]), suggesting that in utero exposure to impaired-nutrition is a risk for obesity and diabetes progression in adulthood.
The BPA-substitute BPS was reported to alter the placental expression of P-glycoprotein (P-gp), one of the main efflux transporters to xenobiotics encoded by the ABCB1 gene
- Conclusion and perspective on enriched environments
This review synthetizes epidemiological studies and experimental data on animals to support the hypothesis that MDs are not only related to our way of eating and moving, but also to early origin and can find their root in DOHaD.
However, if this hypothesis is verified, it will provide new therapeutic strategies to reduce the epidemic of MD. The positive effects of environment stimulation and enrichment have been studied for many years. The concept of “enriched environment” has been proposed in the 40s by Hebb [172]. If prevention strategies failed, it would be interesting to identify therapeutic strategies to avoid or at least dampen negative effects of ELAE on development of MD at adulthood. Robust effects of environmental enrichment in animals suggest that a similar species-typical enrichment of humans could diminish the adverse consequences of urbanization (stress, western diet, lack of physical activity). The relationship between our living environment and human health appears tightly connected. Indeed, lifestyle and environment is not only a social issue, but also biological necessity for public health.
Author Response
We thank the reviewer for his suggestion, effort and involvement in improving our manuscript. We took into account all edit suggested by the review.

Round 2
Reviewer 1 Report
The authors have addressed the comments properly.